

# Improving clinical refractive results of cataract surgery by machine learning

Martin Sramka[1,2], Martin Slovak[2], Jana Tuckova[1] and Pavel Stodulka[2,3]

[1] Department of Circuit Theory/Faculty of Electrical Engineering, Czech Technical University in Prague, Prague, Czech Republic
[2] Research and Development Department, Gemini Eye Clinic, Zlin, Czech Republic
[3] Department of Ophthalmology/Third Faculty of Medicine, Charles University, Prague, Czech Republic

## ABSTRACT

**Aim:** To evaluate the potential of the Support Vector Machine Regression model (SVM-RM) and Multilayer Neural Network Ensemble model (MLNN-EM) to improve the intraocular lens (IOL) power calculation for clinical workflow.

**Background:** Current IOL power calculation methods are limited in their accuracy with the possibility of decreased accuracy especially in eyes with an unusual ocular dimension. In case of an improperly calculated power of the IOL in cataract or refractive lens replacement surgery there is a risk of re-operation or further refractive correction. This may create potential complications and discomfort for the patient.

**Methods:** A dataset containing information about 2,194 eyes was obtained using data mining process from the Electronic Health Record (EHR) system database of the Gemini Eye Clinic. The dataset was optimized and split into the selection set (used in the design for models and training), and the verification set (used in the evaluation). The set of mean prediction errors (PEs) and the distribution of predicted refractive errors were evaluated for both models and clinical results (CR).

**Results:** Both models performed significantly better for the majority of the evaluated parameters compared with the CR. There was no significant difference between both evaluated models. In the ±0.50 D PE category both SVM-RM and MLNN-EM were slightly better than the Barrett Universal II formula, which is often presented as the most accurate calculation formula.

**Conclusion:** In comparison to the current clinical method, both SVM-RM and MLNN-EM have achieved significantly better results in IOL calculations and therefore have a strong potential to improve clinical cataract refractive outcomes.

Corresponding author
Martin Sramka, sramka@gemini.cz

## INTRODUCTION

Cataract surgery is the principal lens replacement refractive surgical procedure performed in adults and is one of the most commonly performed surgical procedures today (*Abell & Vote, 2014*; *Frampton et al., 2014*; *Wang et al., 2017b*). Every year, more than 11 million eyes undergo intraocular lens (IOL) implantation worldwide. The World

Health Organization has estimated that the number of people blinded by cataracts will increase from 10 million in 2010 to 40 million in 2025, as the population grows (*Pascolini & Mariotti, 2012*). Phacoemulsification with IOL implantation is currently the most common method of treating cataracts and many refractive vision errors for which other conventional methods are not suitable (*Linebarger et al., 1999*). The ultimate goal is to achieve complete postoperative independence of ocular correction. Since significant developments have been made in cataract and refractive surgeries over the past 20 years we are now even closer to meeting this target, although there are still areas in which improvements can be made.

The quality of the patient's post-operative vision depends on an accurate selection of the IOL optical power, which influences the residual post-operative refraction. Improving the refractive result of cataract surgery is a challenge for IOL manufacturers, as are determining accurate methods to calculate a suitable IOL lens power.

To achieve an accurate IOL calculation, a series of scientific and therapeutic approaches are required. These include a thorough examination to determine the reason for the vision loss (*Yamaguchi, Negishi & Tsubota, 2011*), preoperative ocular surface preparation, patient visual preferences, eye biometric measurements (*Astbury & Ramamurthy, 2006*; *Shammas & Shammas, 2015*), precise eye surgery and IOL positioning (*Thulasi, Khandelwal & Randleman, 2016*), and an accurate IOL power calculation method (*Norrby, 2008*; *Lee et al., 2015*).

To determine the optimal IOL power, the calculation formulas are used. These formulas use data from preoperative measurements, examinations, and IOL parameters, all of which may influence the overall outcome. The calculation formulas can be divided into Refraction, Regression, Vergence, Artificial Intelligence and Ray Tracing categories based on their calculation method (*Koch et al., 2017*).

Currently, the most commonly used formulas are from the Vergence formula category, although their accuracy is only able to achieve a ±0.5 diopter (D) from the intended target refraction in 60–80% of eyes (*Melles, Holladay & Chang, 2018*). Their accuracy decreases even further in eyes with non-standard biometric features such as eyes with short or long axial lengths (ALs) (*Abulafia et al., 2015*; *Shrivastava et al., 2018*).

The only currently used IOL calculation approach using Artificial Intelligence is the Hill-RBF formula, which has a reported accuracy of 91% of eyes within a ±0.5 D range from the intended target of refraction (*Haag-Streit AG Koeniz, Switzerland, 2017*). However, there is a number of papers that indicate that Hill-RBF accuracy is not significantly different from the Vergence formula category (*Kane et al., 2017*; *Roberts et al., 2018*; *Shajari et al., 2018*). Unfortunately, there is no publication of the Hill-RBF principle in any peer-reviewed scientific journal, so the only information about the principle itself can be obtained from freely available resources on the Internet. Based on this information, it is possible to find out that the Hill-RBF core is a Radial Basis Function and that the algorithm was trained on the data of more than 12,000 eyes. There is no information of what specific machine learning method is used (*Hill, 2018*; *Snyder, 2019*; *The American Society of Cataract and Refractive Surgery, 2018*; *Haag-Streit AG Koeniz, Switzerland, 2017*). Radial basis functions are used in many applications in the field of biomedical engineering (*Le & Ou, 2016a*, *2016b*).

This paper aims to describe the methodology for selecting and optimizing a dataset for SVM-RM and MLNN-EM training, to describe a methodology for evaluating the accuracy of the model, to evaluate SVM-RM and MLNN-EM for IOL power prediction and to compare the accuracy of both models with the current calculation method used in clinical practice.

A support vector machine (SVM) is a supervised machine learning method serving mainly for classification and, in our case, for regression analysis. The aim of this algorithm is to find a hyperplane that optimally splits the feature space so that training data belonging to different classes lie in the separable spaces (*Smola & Schölkopf, 2004*). To find such a hyperplane on non-linear data, a kernel trick is used to transform data from the original feature space into a higher dimension space where it is already linearly separable (*Herbrich, 1999*; *Jap, Stöttinger & Bhasin, 2015*). SVM regression introduces an epsilon-insensitive loss function that is taken into account when minimizing the error through hyperplane optimization. SVM find their application for example in the field of financial forecasting (*Trafalis & Ince, 2000*), travel time prediction (*Wu et al., 2003*), flood forecasting (*Yu, Chen & Chang, 2006*) and genetics (*Le et al., 2019*).

Multilayer neural networks (MLNN) are known for their exceptional ability to approximate continuous functions (*Mongillo, 2011*; *Wu et al., 2012*) and have been widely used in function approximation, prediction and classification (*Park & Sandberg, 1991*; *Girosi, 1992*; *Clarke & Burmeister, 1997*; *Ferrari & Stengel, 2005*).

The MLNN consists of a collection of inputs and processing units known as neurons which are organized in the network layers. Neuron parameters are set up by the training process described by *Kurban & Beşdok (2009)*. The training process is determined by minimizing an error function that measures the degree of success in the recognition of a given number of training patterns (*Lampariello & Sciandrone, 2001*).

## MATERIALS AND METHODS

The project work can be organized into three main parts: Dataset preparation, model design & training and evaluation (Fig. 1).

Data preparation focuses on the methods used in data collection, storage in the EHR database, data mining, and cleaning and optimization in order to obtain a suitable dataset for training and evaluation. Incorrect integration of these processes could lead to a degradation of data sources and a distortion of the quality of results.

Model design and training focuses on the set-up of suitable SVM-RM and MLNN-EM and their training approach when using the dataset.

Evaluation describes the outcome measures and how the data was analyzed.

This study used the data of patients who underwent cataract or lens replacement surgery from December 2014 to November 2018, at Gemini Eye Clinic, Czech Republic. This study was approved by the Institutional Ethics Committee of the Gemini Eye Clinic (IRB approval number 2019-04) and adhered to the tenets of the Declaration of Helsinki. As this was an anonymous retrospective data collection study, the need for written patient consent was waived by the IRB.
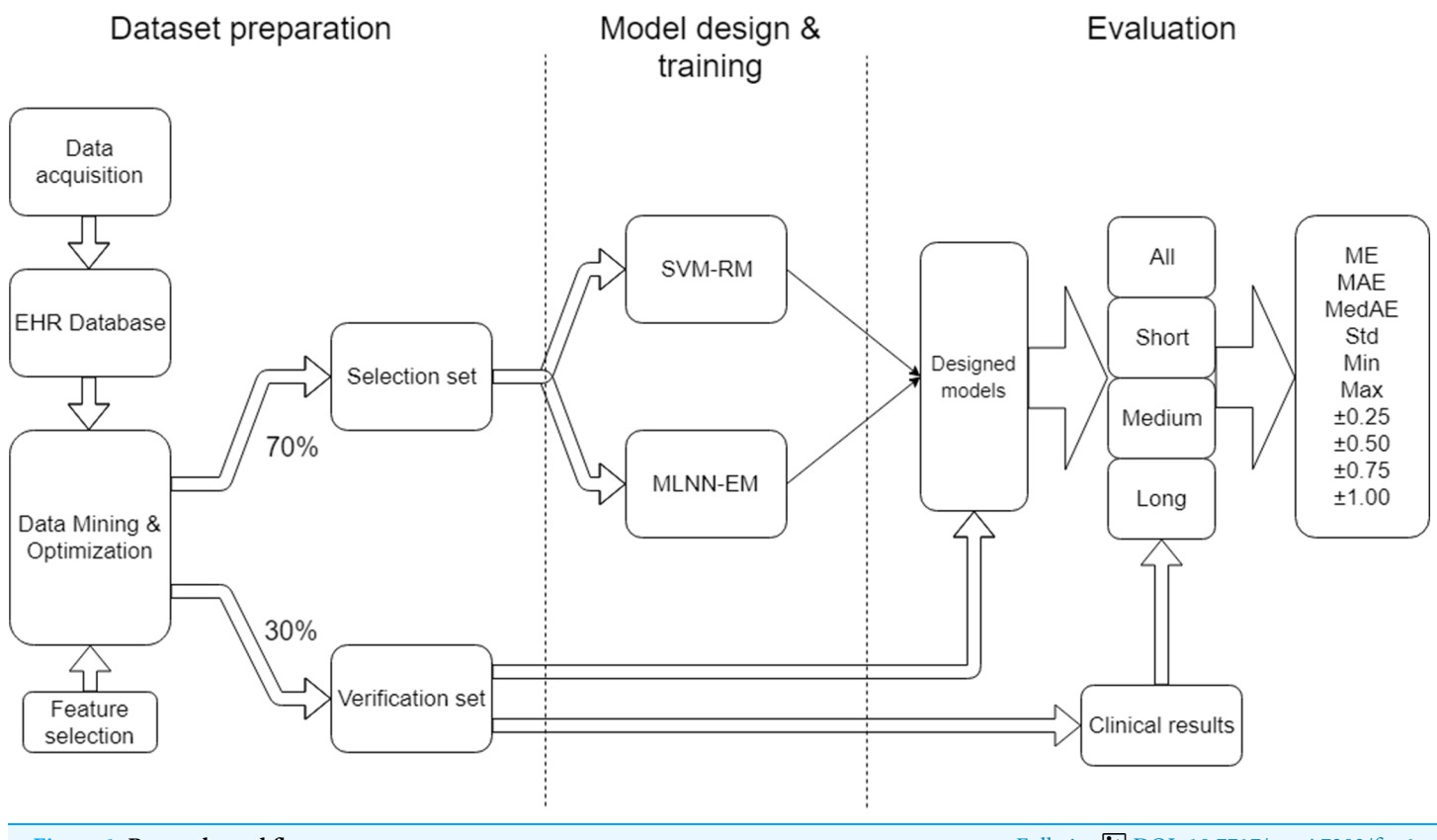

**Figure 1** Research workflow.

## Data acquisition

Data was acquired, recorded and stored by trained staff in the EHR at Gemini Eye Clinic. Data was usually entered before surgery and at follow up visits and post-operative examinations.

The preoperative patient evaluation included distance objective refraction ($Rx_{pre}$), distance subjective refraction, mean keratometry (K), anterior chamber depth (ACD), AL of the eye, uncorrected distance visual acuity (UDVA), corrected distance visual acuity (CDVA), slit lamp examination, retinal examination and contactless intra-ocular pressure examination. Anterior and posterior segment evaluations and biometry measurements were conducted on all patients in the dataset. All biometry examinations (K, ACD, AL) were conducted using the Carl Zeiss IOL Master 500 (Carl Zeiss, Jena, Germany) (*Chen, Hirnschall & Findl, 2011*). All measurements of objective refraction and intraocular pressure were conducted by NIDEK TONOREF II (Nidek, Gamagori, Japan).

All patients in the dataset underwent clear corneal incision surgeries using the Stellaris PC (Bausch and Lomb, Bridgewater, NJ, USA) device. Both continuous curvilinear capsulorhexis (CCC) and IOL implantation in the capsular bag were performed such that the eye was stabilized using an irrigating handpiece introduced into the eye through a side port incision. FineVision Micro F trifocal IOL (Physiol, Lüttich, Belgium) was implanted. All IOLs in the dataset were calculated using the SRK/T formula (*Retzlaff, Sanders & Kraff, 1990*)

with an optimized A constant equal to 119.1. In exceptional cases, the optical power of IOL was corrected based on a decision by the surgeon, especially in the eyes with non-standard biometric specificities. However, all patients' targeted refraction was on emmetropia.

At each follow-up visit, a complete slitlamp evaluation, contactless tonometry, distance objective refraction ($Rx_{post}$), distance subjective refraction, near subjective refraction, keratometry, UDVA, CDVA, uncorrected near visual acuity, and corrected near visual acuity measurements were performed.

The post-operative examination results were collected at least 25 days after surgery, which is the shortest time we consider for sufficient vision recovery based on conclusions made in the work of *Conrad-Hengerer et al. (2015)*.

## Feature selection

Based on the database data integrity, we selected K, ACD, AL, Age, $Rx_{pre}$ as our model input parameters. $Rx_{post}$ and the optical power of implanted IOL ($IOL_{Implanted}$) were used in training target definition. The potential limitation of this selection is further discussed in the discussion section.

## Data mining and optimization

The EHR system data were stored using the SQL Server (Microsoft, Redmond, WA, USA) relational database technology. A single purpose SQL script was designed to get an initial dataview, which was then data mined further in order to obtain a master dataset (MD). The following inclusion and exclusion criteria were used in order to filter the data from physiologically implausible entries and non-standard surgical cases.

Inclusion criteria used to obtain MD:

- ACD between one and five mm
- Preoperative and postoperative UDVA > CDVA in [logMAR]
- AL between 15 and 40 mm
- Mean K between 30 and 60 D
- Patient age between 18 and 99
- Optical power of implanted IOL between six and 35 D.

Examinations and values excluded from the MD for each eye in case of:

- Non-standard surgical procedure or intraoperative complications or any complications affecting postoperative vision recovery

    ○ Surgery record contained any of the strings: "ruptura", "fenestrum", "vitrektom", "praskl", "sklivec", "prolaps", "explant", "sulc", "sulk", "rzp", "key hole"

- Had ocular disease or any corneal pathology

    ○ Patient finding record contained any of the strings: "otok", "striat", "edem", "odchlípen", "PEX", "jizv", "amok", "aparát", "defekt", "degener", "endotelopati", "fibrin", "guttat", "haze", "hemoftalm", "hemophtalm", "luxov", "membrán", "precip", "zonul"

- Previous intraocular surgery or previous corneal refractive surgery

  ○ Patient diagnosis record contains any of strings: "LASIK", "LASEK", "PRK", "LASER", "RELEX", "DMEK", "DALK", "PKP"

- Post-operative CDVA higher than 0.3 logMAR which is widely considered to be a driving standard limit (Visual Standards for Driving in Europe, Consensus paper, European Council of Optometry and Optics)
- Incomplete biometry and refraction measurements
- Preoperative corneal astigmatism of more than 3.0 diopters
- Incomplete EHR documentation
- The difference in AL to second eye >1 mm.

All of the excluded cases, which were identified using strings, came from Czech medical terminology and indicated undesirable contraindications for our application.

All samples containing outliers for K, ACD, AL, Age, $Rx_{pre}$, $Rx_{post}$ were excluded from an MD based on the ±3 sigma rule as these can be considered to be an error in measurement and inappropriate for model training (*Kononenko & Kukar, 2007*; *Leys et al., 2013*).

The principle of preparing data suitable for training is to find the ideal value for the already implanted IOL ($IOL_{Ideal}$). $IOL_{Ideal}$ is considered to be an IOL that will not induce any residual postoperative refraction for the patient's eye or will not deviate from the intended target refraction (for distance vision this was considered as 0 D). For finding such $IOL_{Ideal}$, the following information is needed:

- Optical power of $IOL_{Implanted}$
- Measured residual refraction $Rx_{post}$
- Interrelationship of $Rx_{post}$ and $IOL_{Implanted}$.

It is generally known that 1.0 D of IOL prediction error (PE) produces approximately 0.7 D of refractive PE at the spectacle plane (*Wang et al., 2017a*). However, this is a general assumption and since the eye is a complex optical system it may not be sufficiently accurate in all eyes. The interrelationship between $Rx_{post}$ and $IOL_{Implanted}$ should also consider eye biometrical parameters representative of the eye optical system, such as the eye AL and the power of the cornea K. The interrelationship of these two variables was determined by reversed Eye Vergence Formula Eq. (1) (*Olsen, 2007*; *Gatinel, 2018*).

$$Rx_{theorPost} = \cfrac{1}{\frac{V}{1,000} - \cfrac{1}{1,000 * \left(\frac{K}{1,000} - 1/1,000 * \left(\frac{ELP}{1,336} - 1/\left(1,336 * \left(\frac{IOL}{1,336} - \frac{1}{AL - ELP}\right)\right)\right)\right)}} \tag{1}$$

Equation 1. Reversed Eye Vergence Formula.

$Rx_{theorPost}$ is the calculated refraction for the eye with specific K in (D), AL in (mm), V (vertex distance) in (mm), IOL in (D) and effective lens position (ELP) in (mm) calculated using recommendations by *Retzlaff, Sanders & Kraff (1990)*.

**Table 1 Selection set population characteristics.**

|  | Mean | Median | Std | Min | Max | $P_{SW}$ | $P_{DP}$ |
|---|---|---|---|---|---|---|---|
| Age (years) | 56.89 | 57.00 | 7.25 | 36.00 | 78.00 | 8.543e-5 | 0.091 |
| K (D) | 43.27 | 43.25 | 1.40 | 39.39 | 47.51 | 0.252 | 0.547 |
| ACD (mm) | 3.10 | 3.10 | 0.32 | 2.21 | 4.10 | 0.189 | 0.350 |
| AL (mm) | 23.03 | 23.07 | 0.92 | 19.94 | 26.26 | 0.010 | 0.111 |
| $Rx_{pre}$ (D) | 1.85 | 1.88 | 1.52 | −3.88 | 6.63 | 0.000 | 0.000 |
| $IOL_{Ideal}$ (D) | 22.80 | 22.50 | 2.74 | 12.62 | 34.17 | 8.615e-12 | 9.992e-16 |

**Note:**
Standard deviation (Std), Minimum (Min), Maximum (Max), Shapiro–Wilk *P*-value (pSW) and D'Agostino-Pearson's K2 *P*-value (pDP). Selection set was assessed for normality by Shapiro–Wilk and D'Agostino-Pearson's K2 normality tests at level of $P = 0.001$.

A change of the refraction at spectacle plane by changing the IOL power value was computed using Eq. (2), and the $IOL_{Ideal}$ calculation is expressed by Eq. (3)

$$Rx_{05IOL} = Rx_{theorPost}(IOL) - Rx_{theorPost}(IOL + 0.5) \qquad (2)$$

Equation 2. Dioptric change of refraction at spectacle plane in case of IOL value change of 0.5 (D).

$$IOL_{Ideal} = IOL_{Implanted} + \left( \frac{Rx_{post}}{Rx_{05IOL}} \right) * 0.5 \qquad (3)$$

Equation 3. Calculation of ideal value of IOL for the specific eye.

MD was then randomly divided into the selection set and the verification set in a proportion of 70% to 30%. Selection set variables were normalized using the mapminmax MATLAB 2017a (MathWorks, Natick, MA, USA) routine, which maps row minimal and maximal values between −1 and 1. Every verification set variable was cleared from samples out of the minimum and maximum range of the selection set to avoid the PE on non-trained data. Verification set variables were then normalized using mapminmax with the same normalization parameters.

## Data description

The selection set (70% of MD, Table 1) contained information from 1,539 eyes (771 right eyes, 768 left eyes) of 1,168 patients (540 male, 628 female).

Age failed in a normality assessment by Shapiro–Wilk ($P_{SW}$; Table 1) but was confirmed by D'Agostino-Pearson's K2 test ($P_{DP}$; Table 1). $Rx_{pre}$ (Fig. 2A) and $IOL_{Ideal}$ (Fig. 2B) failed in a normality assessment by both normality tests.

The verification set (30% of MD, Table 2) contained information from 655 eyes (340 right eyes, 315 left eyes) of 591 patients (272 male, 319 female). As in the selection set case, only $Rx_{pre}$ (Fig. 2C) and $IOL_{Ideal}$ (Fig. 2D) failed in normality assessment by both normality tests ($P_{SW}$, $P_{DP}$; Table 2).

## Machine learning

The selection set was used to find the design and training of each model. The verification set was used to evaluate the results. No samples from the verification set were introduced

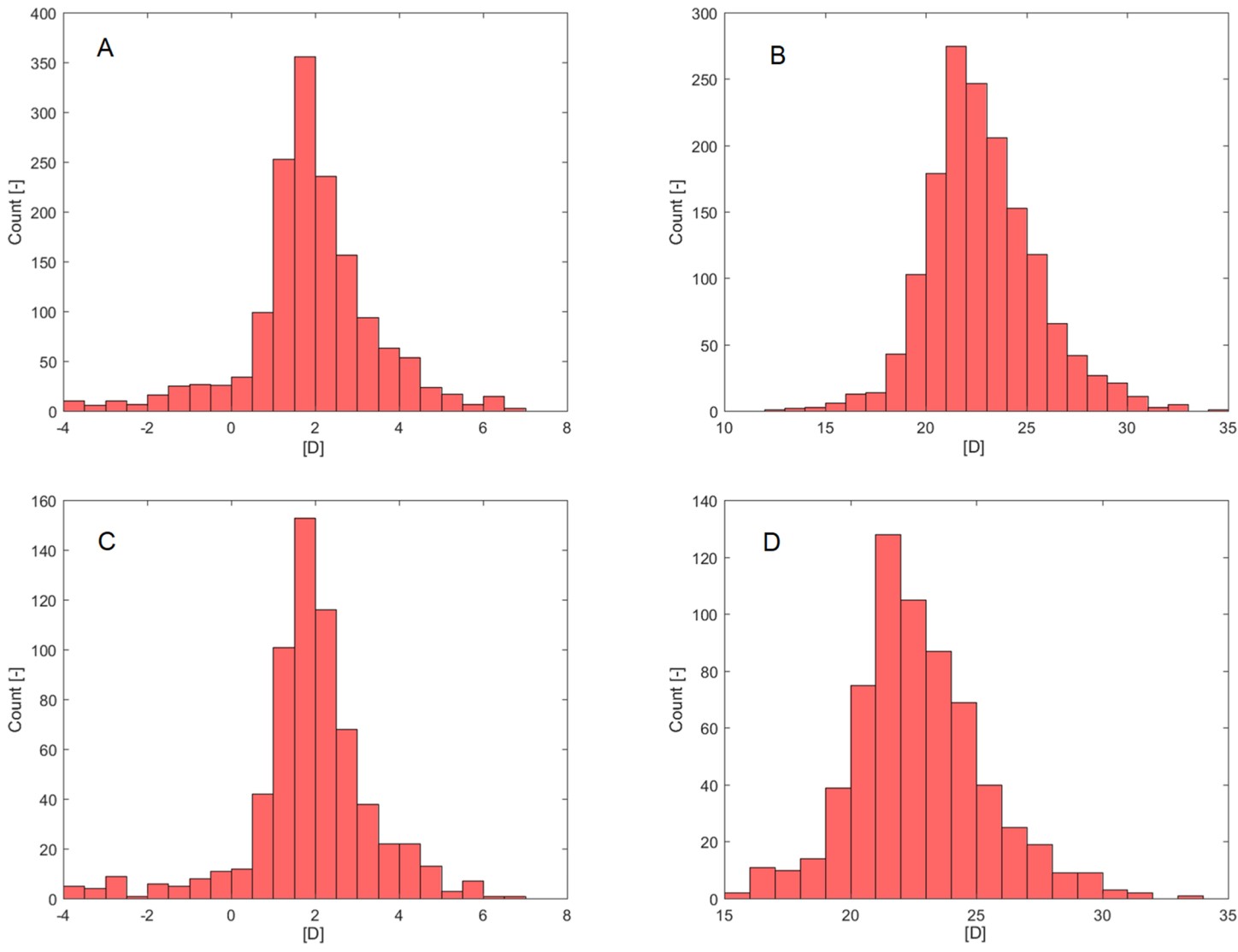

**Figure 2  Histograms.** (A) $Rx_{pre}$—Selection set. (B) $IOL_{Ideal}$—Selection set. (C) $Rx_{pre}$—Verification set. (D) $IOL_{Ideal}$—Verification set.

to the model during the design and training phase, conversely, no samples from the selection set were used for model evaluation. Our model predictors were variables mentioned in the feature selection section K, ACD, AL, Age, $Rx_{pre}$. The training target was $IOL_{Ideal}$ and the prediction outcome was $IOL_{Predicted}$.

## SVM-RM

Our SVM-RM was designed and trained in MATLAB 2017a (MathWorks, Natick, MA, USA) using the fitrsvm (*MathWorks, 2017a*) method. Finding the appropriate hyperparameters for a given task is one of the most important steps in designing the model, and is vital to its accuracy and key for the training and testing times (*Wang & Gong, 2018*). The optimal hyperparameters of the model were found through the optimize

**Table 2 Verification set population characteristics.**

|  | Mean | Median | Std | Min | Max | $P_{SW}$ | $P_{DP}$ |
|---|---|---|---|---|---|---|---|
| Age (years) | 56.83 | 56.00 | 7.29 | 37.00 | 76.00 | 0.003 | 0.161 |
| K (D) | 43.33 | 43.30 | 1.33 | 39.41 | 46.92 | 0.263 | 0.199 |
| ACD (mm) | 3.11 | 3.10 | 0.32 | 2.29 | 4.06 | 0.183 | 0.206 |
| AL (mm) | 23.03 | 22.99 | 0.90 | 20.17 | 25.88 | 0.530 | 0.417 |
| $Rx_{pre}$ (D) | 1.83 | 1.75 | 1.49 | −3.88 | 6.63 | 1.998e-15 | 0 |
| $IOL_{Ideal}$ (D) | 22.71 | 22.42 | 2.64 | 15.32 | 33.51 | 7.793e-7 | 3.467e-7 |

**Note:**
Standard deviation (Std), Minimum (Min), Maximum (Max), Shapiro–Wilk $P$-value (pSW) and D'Agostino-Pearson's K2 $P$-value (pDP).

**Table 3 SVM-RM parameters.**

| Kernel function | Polynomial |
|---|---|
| Kernel scale | – |
| Epsilon | 0.0282 |
| Box constraint | 0.0049 |
| Polynomial order | 2 |
| MSE | 0.0032 |

**Note:**
MSE, Mean squared error.

hyperparameters (*MathWorks, 2017a*) method that searched for optimal kernel function, kernel scale, epsilon, box constraint and polynomial order.

The selection set was used in model training by the sequential minimal optimization algorithm (*Zeng et al., 2008*) with 30% of randomly selected data used for holdout validation. The model parameters are summarized in Table 3.

## MLNN-EM

For the MLNN performance improvement ensemble median was preferred over ensemble averaging reported by *Kourentzes, Barrow & Crone (2014)*.

Our MLNN presented in Fig. 3 was designed and trained in MATLAB 2017a (MathWorks, Natick, MA, USA) by fitnet (*MathWorks, 2017b*). It had one hidden layer with five neurons and one output layer with one neuron with linear transfer function. The internal structure and links of the MLNN are described, for example, by Tuckova or in the MATLAB 2017a documentation (*Tuckova, 2009*; *MathWorks, 2017b*). The Hyperbolic Tangent Sigmoid transfer function was used as a transfer function in the hidden layer and is proposed by many authors as a good choice for multivariate functions approximation (*Anastassiou, 2011*; *Romero Reyes et al., 2013*). The Levenberg–Marquardt backpropagation algorithm was used for model training using the trainlm (*MathWorks, 2017c*) method (*Ranganathan, 2004*).

The ensemble median factor was set to 10 which means that 10 MLNN were trained by the selection set in order to produce a desired output taken as a median of all outputs. Weights and biases were initialized by the Nguyen-Widrow initialization function for each ensemble training cycle (*Nguyen & Widrow, 1990*).
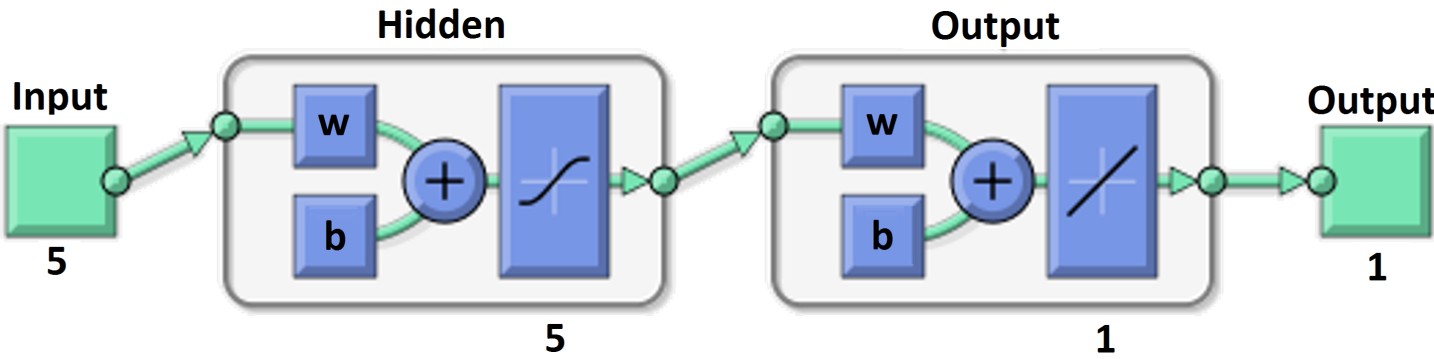

**Figure 3 MLNN layer structure.**

**Table 4 MLNN-EM design parameters.**

|  | Mean | Median | Std | Min | Max |
|---|---|---|---|---|---|
| Train MSE | 0.00302 | 0.00306 | 9.44729E-05 | 0.0028 | 0.00311 |
| Validation MSE | 0.00307 | 0.00310 | 0.00033 | 0.0025 | 0.00364 |
| Test MSE | 0.00329 | 0.00333 | 0.00039 | 0.0025 | 0.00387 |
| Epoch | 22.8 | 21.5 | 18.6 | 7 | 72 |

**Note:**
MSE, Mean squared error.

The early stopping algorithm was used to overcome the model overfitting each ensemble training cycle. The selection set was randomly divided into three groups, for network training, validation, and testing by a 70:15:15 ratio (*Ross et al., 2009*). MLNN training was stopped when the network performance in the validation group failed to improve or remained the same for 20 epochs. The weights and biases at the minimum of the validation error were returned for each ensemble model. Training, validation and test performances for our MLNN-EM are summarized in Table 4.

The optimal number of neurons in the MLNN hidden layer was found iteratively, testing all available combinations of neurons from one up to 350 neurons in a hidden layer. The topology ensemble which ensured the smallest median + 1× standard deviation (STD) of the test mean square error (MSE) was selected for next processing. With the rising number of the neurons in the hidden layer, the test MSE also grew (Fig. 4).

The default values of the MATLAB functions were used, unless otherwise mentioned. All of these parameters can be found in the MATLAB documentation (*MathWorks, 2019*).

## Evaluation methodology and statistical tests

The results predicted by each model were compared against the clinical results (CR) that were achieved and both models were compared with each other. We followed the recommendations described in the work of Wang (*Wang et al., 2017a*) for evaluating the results and in performing statistical analysis. The mean numerical PE, mean absolute PE (MAE), median absolute PE (MedAE), STD, minimum PE, maximum PE, and percentages of eyes within PE targets of ±0.25 D, ±0.50 D, ±0.75 D, ±1.00 D, were determined for $Rx_{post}$ and refraction calculated from $IOL_{Predicted}$ ($Rx_{predicted}$). The $Rx_{predicted}$ calculation is described in Eq. (4).

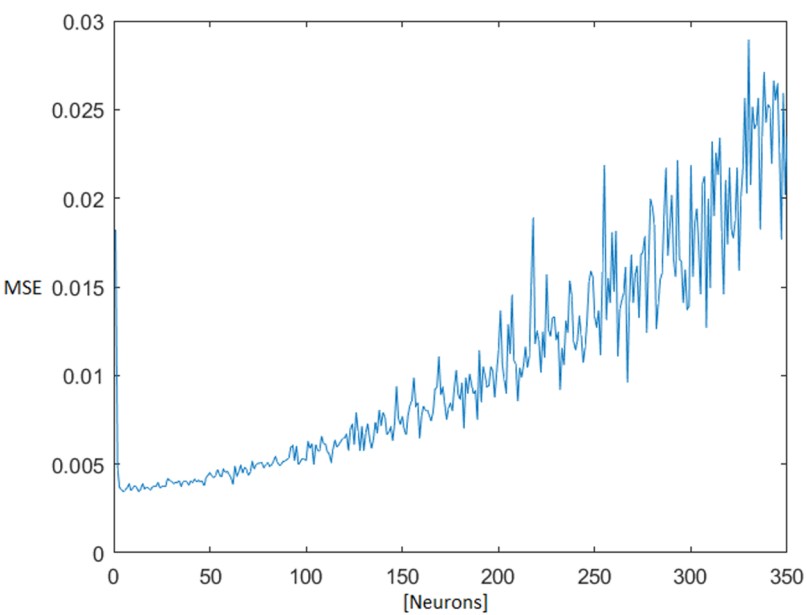

**Figure 4 MSE dependence on the number of neurons in the hidden layer.** Mean square error (MSE).

$$\text{Rx}_\text{predicted} = \left(\frac{\text{IOL}_\text{Implanted} - \text{IOL}_\text{Predicted}}{0.5}\right) * \text{Rx}_\text{05IOL} + \text{Rx}_\text{post} \tag{4}$$

Equation 4. Calculation of $\text{Rx}_\text{predicted}$ from $\text{IOL}_\text{Predicted}$.

Since AL is referred to as being the most important in predicting the IOL power (*Mahdavi & Holladay, 2011*), the evaluation process is usually divided into subgroups based on AL (*Wang et al., 2017a*). The verification set was thus divided into the following AL subgroups:

- SHORT eyes group—eyes with AL <= 22 mm—81 samples
- MEDIUM eyes group—eyes with 22 mm < AL < 24 mm—480 samples
- LONG eyes group—eyes with AL => 24 mm—94 samples
- ALL eyes group—whole verification set with all eyes—655 samples.

Statistical analysis was performed using MATLAB 2017a (MathWorks, Natick, MA, USA).

The Wilcoxon test (*Mercier et al., 2015*) was used to assess the MAE and MedAE differences between the real clinical calculation results and both models. The McNemar test (MN) with Yates' correction (*Westfall, Troendle & Pennello, 2010*) was used to evaluate the difference in the percentage of eyes in a certain PE diopter group found between real clinical calculation results and both models. The MN and Sing test (*Dixon & Mood, 1946*) were added to test both models as well. The Bonferroni correction was applied for multiple comparisons. The level of significance was set at 0.05 and all *P*-values were reported.

**Table 5 Prediction errors in the ALL axial length group for clinical results (CR), SVM-RM and MLNN-EM.**

|  | CR | SVM-RM | MLNN-EM |
|---|---|---|---|
| ME | −0.464 | 0.012 | 0.002 |
| MAE | 0.523 | 0.310 | 0.309 |
| MedAE | 0.500 | 0.260 | 0.258 |
| Std | 0.433 | 0.395 | 0.395 |
| Min | −1.875 | −1.480 | −1.514 |
| Max | 1.125 | 1.372 | 1.310 |
| Eyes within PE (%) |  |  |  |
| ±0.25 | 33.4 | 48.2 | 48.9 |
| ±0.50 | 57.7 | 82.8 | 82.3 |
| ±0.75 | 79.4 | 93.4 | 93.7 |
| ±1.00 | 91.8 | 97.7 | 97.7 |

Note:
Mean prediction error (ME), Mean absolute prediction error (MAE), Median absolute prediction error (MedAE), Standard deviation (Std), Minimum prediction error (Min), Maximum prediction error (Max), Prediction error (PE).

## RESULTS

Table 5 shows the results for all evaluated parameters in the ALL AL sample group. In comparison to CR, both models showed significantly lower absolute error (SVM-RM $P = 3.422e-78$ and MLNN-EM $P = 2.841e-76$). MLNN-EM had a lower absolute error than SVM-RM but this was not statistically significant. The overall percentage of eyes with PEs between ±0.25, ±0.50, ±0.75 and ±1.00 D compared to CR was significantly higher for both models (SVM-RM $P_{\pm0.25} = 7.860e-7$, $P_{\pm0.50} = 0$, $P_{\pm0.75} = 1.443e-15$, $P_{\pm1.00} = 4.823e-7$ and MLNN-EM $P_{\pm0.25} = 2.140e-7$, $P_{\pm0.50} = 0$, $P_{\pm0.75} = 1.110e-16$, $P_{\pm1.00} = 2.992e-7$). MLNN-EM performed better than SVM-RM in ±0.25 D, ±0.75 D and worse or the same for ±0.50 D, ±1.00 D PE groups but this was not statistically significant.

Table 6 shows the results for all evaluated parameters in the SHORT AL sample group. Compared to CR, both models had significantly lower absolute error (SVM-RM $P = 3.674e-7$ and MLNN-EM $P = 7.445e-8$), SVM-RM performed significantly better for ±0.50 D and ±1.00 D PE groups ($P_{\pm0.50} = 0.029$ and $P_{\pm1.00} = 0.041$) and better for ±0.25 D and ±0.75 D PE groups ($P_{\pm0.25} = 0.735$ and $P_{\pm0.75} = 0.070$) but this was not statistically significant, MLNN-EM performed significantly better for ±0.50 D and ±0.75 D ($P_{\pm0.50} = 0.046$ and $P_{\pm0.75} = 0.027$) and worse for ±0.25 D and ±1.00 D PE groups ($P_{\pm0.25} = 0.429$ and $P_{\pm1.00} = 0.131$), but this was not statistically significant. MLNN-EM had a lower absolute error than SVM-RM but this was not significant. MLNN-EM performed better than SVM-RM in ±0.25 D and ±0.75 D PE groups and worse or the same for ±0.50 D and ±1.00 D PE groups but this was not statistically significant.

Table 7 shows the results for all evaluated parameters in the MEDIUM AL sample group. Compared to CR, both models had significantly lower absolute error (SVM-RM $P = 3.674e-7$ and MLNN-EM $P = 7.445e-8$), and both SVM-RM and MLNN-EM performed significantly better for all PE groups (SVM-RM $P_{\pm0.25} = 5.699e-6$, $P_{\pm0.50} = 0$, $P_{\pm0.75} = 1.257e-10$, $P_{\pm1.00} = 1.009e-3$ and MLNN-EM $P_{\pm0.25} = 3.595e-6$, $P_{\pm0.50} = 0$, $P_{\pm0.75} = 2.025e-11$,

**Table 6  Prediction errors in the SHORT axial length group for Clinical Results (CR), SVM-RM and MLNN-EM.**

|  | CR | SVM-RM | MLNN-EM |
|---|---|---|---|
| ME | −0.369 | 0.002 | 0.018 |
| MAE | 0.465 | 0.322 | 0.320 |
| MedAE | 0.500 | 0.302 | 0.266 |
| Std | 0.464 | 0.399 | 0.398 |
| Min | −1.500 | −0.865 | −0.930 |
| Max | 1.125 | 0.929 | 1.007 |
| Eyes within PE (%) |  |  |  |
| ±0.25 | 40.7 | 44.4 | 48.1 |
| ±0.50 | 63.0 | 76.5 | 76.5 |
| ±0.75 | 85.2 | 93.8 | 95.1 |
| ±1.00 | 92.6 | 100.0 | 98.8 |

Note:
Mean prediction error (ME), Mean absolute prediction error (MAE), Median absolute prediction error (MedAE), Standard deviation (Std), Minimum prediction error (Min), Maximum prediction error (Max), Prediction error (PE).

**Table 7  Prediction errors in the MEDIUM axial length group for clinical results (CR), SVM-RM and MLNN-EM.**

|  | CR | SVM-RM | MLNN-EM |
|---|---|---|---|
| ME | −0.466 | 0.024 | 0.008 |
| MAE | 0.523 | 0.307 | 0.307 |
| MedAE | 0.500 | 0.251 | 0.254 |
| Std | 0.424 | 0.396 | 0.395 |
| Min | −1.875 | −1.480 | −1.514 |
| Max | 0.875 | 1.372 | 1.310 |
| Eyes within PE (%) |  |  |  |
| ±0.25 | 33.1 | 49.6 | 49.4 |
| ±0.50 | 56.9 | 83.8 | 82.9 |
| ±0.75 | 79.8 | 93.3 | 93.5 |
| ±1.00 | 92.9 | 97.3 | 97.5 |

Note:
Mean prediction error (ME), Mean absolute prediction error (MAE), Median absolute prediction error (MedAE), Standard deviation (Std), Minimum prediction error (Min), Maximum prediction error (Max), Prediction error (PE).

$P_{\pm1.00} = 3.164e\text{-}4$). MLNN-EM had a lower absolute error than SVM-RM, but this was not significant. MLNN-EM performed better than SVM-RM in ±0.75 D and ±1.0 D PE groups and worse for ±0.25 D and ±0.50 D PE groups but this was not statistically significant.

Table 8 shows the results for all evaluated parameters in the LONG AL sample group. Compared to CR, both models had significantly lower absolute error (SVM-RM $P = 3.954e\text{-}13$ and MLNN-EM $P = 1.289e\text{-}13$), and both SVM-RM and MLNN-EM performed significantly better for all PE groups (SVM-RM $P_{\pm0.25} = 0.041$, $P_{\pm0.50} = 4.785e\text{-}5$, $P_{\pm0.75} = 2.152e\text{-}5$, $P_{\pm1.00} = 3.283e\text{-}3$ and MLNN-EM $P_{\pm0.25} = 0.030$, $P_{\pm0.50} = 4.976e\text{-}5$, $P_{\pm0.75} = 2.151e\text{-}5$, $P_{\pm1.00} = 3.283e\text{-}3$). MLNN-EM had a lower absolute error than SVM-RM but this was not significant. MLNN-EM performed better than SVM-RM in ±0.25 D and

**Table 8  Prediction errors in the LONG axial length group for clinical results (CR), SVM-RM and MLNN-EM.**

|  | CR | SVM-RM | MLNN-EM |
|---|---|---|---|
| ME | −0.535 | −0.043 | −0.043 |
| MAE | 0.574 | 0.316 | 0.311 |
| MedAE | 0.500 | 0.270 | 0.269 |
| Std | 0.442 | 0.387 | 0.393 |
| Min | −1.625 | −1.013 | −1.000 |
| Max | 0.875 | 1.096 | 1.230 |
| Eyes within PE (%) |  |  |  |
| ±0.25 | 28.7 | 44.7 | 46.8 |
| ±0.50 | 57.4 | 83.0 | 84.0 |
| ±0.75 | 72.3 | 93.6 | 93.6 |
| ±1.00 | 85.1 | 97.9 | 97.9 |

Note:
Mean prediction error (ME), Mean absolute prediction error (MAE), Median absolute prediction error (MedAE), Standard deviation (Std), Minimum prediction error (Min), Maximum prediction error (Max), Prediction error (PE).

±0.50 D PE groups, but this was not statistically significant and same for ±0.75 D and ±1.00 D PE groups.

*P*-values for mutual evaluation of both models are presented in Table 9. For clarity, a chart comparing PE of all groups is presented in Fig. 5.

## DISCUSSION

We have described the methodology of selecting and optimizing the dataset for SVM-RM and MLNN-EM training, and compared the accuracy of both models with the current calculation method used in clinical practice. Overall, the percentages of eyes with PEs between ±0.25, ±0.50, ±0.75 and ±1.00 D for both models were significantly better for the vast majority of evaluated parameters when compared to CR. Insignificant improvement occurred only in PE ± 0.25 D and ±0.75 D groups for the SHORT AL subset. As previously mentioned, calculations for eyes with a short AL are more problematic due to the more complex ELP prediction and because of the higher probability of a steep cornea and a shallow ACD (*Hoffer, 1980*). Compared with CR, both models in all AL subgroups had a smaller SD, which expresses a higher certainty of the calculation method (*Shajari et al., 2018*). Long eyes over 26.3 mm and extremely long eyes were not included in this study.

Compared to the results of the Barrett Universal II formula obtained from the literature (Table 10), which is often presented as the most accurate calculation formula, the accuracy achieved by SVM-RM and MLNN-EM is competitive (*Cooke & Cooke, 2016*; *Kane et al., 2016*, *2017*; *Shajari et al., 2018*), and the results achieved in the ±0.50 D PE category with SVM-RM and MLNN-EM were even slightly better. However, in order to objectively compare the results, it would be necessary to evaluate all methods on the same datasets and not rely solely on the outcomes source found in the literature.

Mutual evaluation did not show a significant difference between the tested models so it can be said that both provide similar accuracy of the calculations in all tested PE groups.

Table 9  Mutual evaluation of difference between SVM-RM and MLNN-EM.

|  | ALL | SHORT | MEDIUM | LONG |
|---|---|---|---|---|
| PE WT | 0.679 | 0.763 | 0.545 | 0.917 |
| ±0.25 MN | 0.819 | 0.449 | 0.891 | 0.802 |
| ±0.50 MN | 0.735 | 0.723 | 0.540 | 1 |
| ±0.75 MN | 0.789 | 1 | 1 | 0 |
| ±1.00 MN | 0.723 | 1 | 1 | 0.479 |
| ±0.25 ST | 0.819 | 0.453 | 0.891 | 0.803 |
| ±0.50 ST | 0.735 | 1 | 0.541 | 1 |
| ±0.75 ST | 0.790 | 1 | 1 | 1 |
| ±1.00 ST | 1 | 1 | 1 | 1 |

Note:
Absolute prediction error (PE) by Wilcoxon test (WT), McNemar test (MN), Sign test (ST).

Both model predictions were almost identical and when compared to the CR there was slightly larger, minimal error, which we were not expecting.

Undoubtedly, the reason for the significantly worse results of the CR group is its simplicity, where only AL and K are used for the IOL power calculation. In order to increase the calculation accuracy, modern calculation methods take into account more circumstances, which could affect the refractive predictability of the surgery (Olsen, 2007; Haigis, 2012; Gökce et al., 2018). Input parameters used in our models are standard parameters acquired from regular patient examination prior to cataract surgery. Thus, it does not introduce any additional requirements on the data acquisition.

The poorer results from the CR group could also be due to the non-optimized constant of the implanted IOL. This is seen in the mean error of the CR group, which has a range between −0.369 and −0.535 D among all AL subsets. Our method of $IOL_{Ideal}$ calculation optimizes the mean error of prediction to zero. This mechanism of $IOL_{Ideal}$ calculation can thus influence the mean error based on the desired refraction.

Table 11 describes the input parameters used by contemporary formulas (Olsen, 2007). Our model input parameters are K, ACD, AL, Age and $Rx_{pre}$, which are all of the possible calculation variables which could be utilized during the data mining process.

The IOL Master 500 used in the biometry examination to gather the anatomical data is not able to measure lens-thickness. However the influence on the precision could probably be overlooked as it is said to be the second least important calculation factor (Gale et al., 2009). Conversely, it can have a greater influence on the IOL calculation than K (Olsen, 2006). Another way to improve the accuracy of the calculations would be to find out how to extract information from incomplete white to white measurements as this value is referred to as the third most important in predicting ELP (Mahdavi & Holladay, 2011). It is possible to determine how to handle missing values in datasets in order to maximize the information gain (Kaiser, 2014).

In order to avoid the distortion of statistical analysis by correlated data, it is recommended to include only one eye per patient in analyses (Armstrong, 2013). Our verification set contained less than 10% of the data that came from both eyes of patients. This means that the intra-class correlation factor would be less than 0.1 in the worst

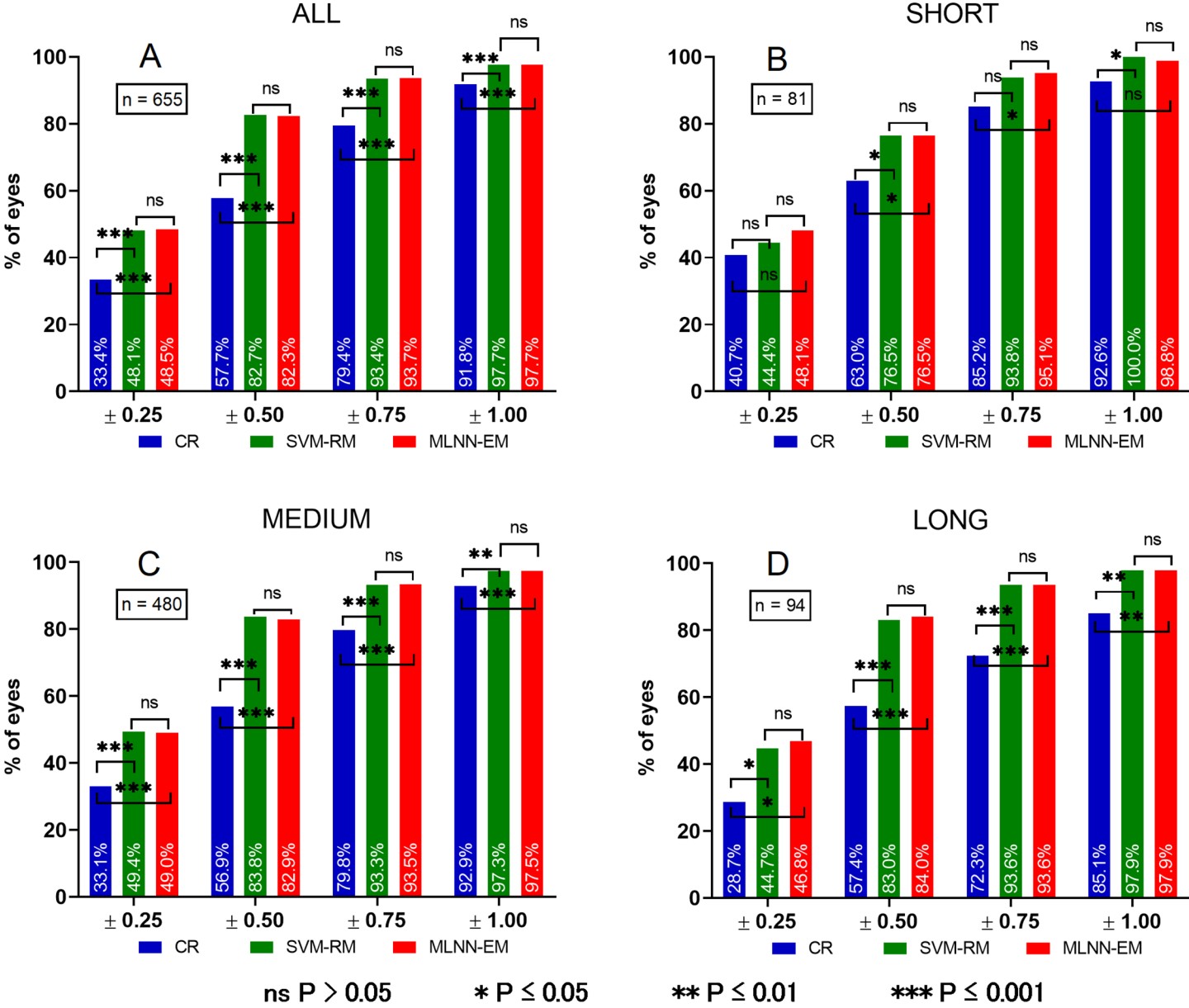

**Figure 5 Histograms of PE in different eye AL groups.** Prediction error (PE). (A) Prediction error in ALL eyes group, (B) prediction error in SHORT eyes group, (C) prediction error in MEDIUM eyes group, (D) prediction error in LONG eyes group. ns $P > 0.05$, *$P \leq 0.05$, **$P \leq 0.01$, ***$P \leq 0.001$.

scenario (the between eye correlation equals one—for every applicable patient in the verification set) indicating an extremely poor correlation (*Cicchetti, 1994*; *Koo & Li, 2016*). We have thus concluded that it is safe to use conventional methods of statistical analysis while including the maximum number of eyes in our datasets.

Our method does not use A constants, as usual formulas, so both models are designed as lens-specific and the ELP prediction is coded directly into the internal structures of the model. The IOL power calculation for another IOL would require going through the entire process of data preparation, model design, and training and evaluation.

**Table 10 Prediction error comparison for Barrett Universal II, SVM-RM and MLNN-EM for all axial lengths.**

| Eyes within PE (%) | Barrett Universal II | SVM-RM | MLNN-EM |
|---|---|---|---|
| ±0.25 | 43.5–60.0 | 48.1 | 48.5 |
| ±0.50 | 72.3–80.6 | 82.7 | 82.3 |
| ±1.00 | 94.5–99.7 | 97.7 | 97.7 |

Note:
Prediction error (PE).

**Table 11 Overview of contemporary formulas input parameters.**

|  | Hill-RBF | HofferQ | Holladay 1 | Holladay 2 | SRK/T | Haigis | Olsen |
|---|---|---|---|---|---|---|---|
| K | x | x | x | x | x | x | x |
| AL | x | x | x | x | x | x | x |
| ACD | x |  |  | x |  | x | x |
| LT |  |  |  | x |  |  | x |
| WTW | x |  |  | x |  |  | x |
| Age |  |  |  | x |  |  |  |
| Rx-pre |  |  |  | x |  |  |  |

Note:
K, mean keratometry; AL, axial length; ACD, anterior chamber depth; LT, lens thickness; WTW, white to white; Age, patients age; Rx-pre, preoperative refraction.

However, due to the fact that there are many small datasets machine learning strategies, it would not be necessary to search for the same amount of training data (*Jiang, Li & Zhou, 2009*; *Olson, Wyner & Berk, 2018*). The final limitation may be the unknown training accuracy outside the input variables training range.

## CONCLUSIONS

This study indicated that SVM-RM and MLNN-EM have a strong potential for improving clinical IOL calculations. The greater optimization and accuracy of IOL calculations reduces the risk of subsequent reoperation or potential refractive laser corrections and the associated risk of complications as well as improving comfort for the patient.

Additional research will be focused on testing the next machine learning algorithms that might be suitable for IOL calculations such as convolutional neural networks, which are mainly used in image processing but more often in the field of biomedical engineering (*Le, Ho & Ou, 2017*, *2018*; *Le & Nguyen, 2019*) as well as on the implementation of both models to our EHR system.

## ACKNOWLEDGEMENTS

This work is a summary of independent research. The authors would like to thank the reviewers for their valuable suggestions.

### Funding

The authors received no funding for this work.

## Competing Interests

Martin Sramka, Martin Slovak and Pavel Stodulka are employed by the Gemini Eye Clinic.

## Author Contributions

- Martin Sramka conceived and designed the experiments, performed the experiments, analyzed the data, prepared figures and/or tables, authored or reviewed drafts of the paper, approved the final draft.
- Martin Slovak analyzed the data, prepared figures and/or tables, authored or reviewed drafts of the paper, approved the final draft.
- Jana Tuckova authored or reviewed drafts of the paper, approved the final draft, artificial Intelligence methods supervisor.
- Pavel Stodulka authored or reviewed drafts of the paper, approved the final draft, ophthalmology methods supervisor.

## Ethics

The following information was supplied relating to ethical approvals (i.e., approving body and any reference numbers):

The Institutional Ethics Committee of the Gemini Eye Clinic approved this study (IRB approval numer 2019-04).

## Data Availability

Raw data and source codes are available at GitHub (martymaker164/ML_IOL): https://github.com/martymaker164/ML_IOL/tree/master/Algo.

Raw data are saved in IOL_Report_ANN.mat in folders "ANN-EM" and "SVM-RM."

## Supplemental Information

Supplemental information for this article can be found online at http://dx.doi.org/10.7717/peerj.7202#supplemental-information.

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
