# Peer review of "Improving clinical refractive results of cataract surgery by machine learning"

_PeerJ, doi:10.7717/peerj.7202_

## Round 0.1 · original submission · Major Revisions

Dear authors,
as you can see below, both Referees asked for a major revision, and I concur with that. There are several topics that need some attention.

- The language can be improved. As an example, the abstract: "A dataset containing 2194 eyes". Well, I guess the data set does not physically contain eyes, right? I would say, information about 2194 eyes, eye interventions, etc.
I suggest you, as a minimum, to use some grammar checker, like Word's one.

- Section data description. It's a good practice to report the obtained p-value, and not just the result of the statistical test.
In this same section, what does "histogram analysis" mean? I don't see much normality in the histograms, there are too many extreme values. That's why it is good to use statistical tests: the naked eye can be fooled!

- I agree with the second Referee, in that Figs 2-5 can be merged into a single one.

- Finally, I understand why the validation is divided in three groups: short, medium and long. Yet, the models are trained with all the data, right? Wouldn't it be better to train specific models for each one of these groups? The result may drastically improve, as maybe each group has different characteristics.

Reviewer 1 ·

Basic reporting

Some sentences are hard to follow. English editing may help.

Experimental design

Excellent, although some improvements are desirable.

Validity of the findings

Improvements are needed.

Additional comments

Based on machine learning, this study evaluated two models for IOL power calculation. It is an interesting study. However, the following comments should be addressed:

1. Lines 297-298: It states “The results predicted by each model were compared against the achieved “clinical results” (CR). Figures 5-8 showed that the mean errors for the CR were about -0.4 to -0.5 D, whereas the mean errors for the two machine models were close to zero. The IOL lens constant should be optimized first in order to achieve a mean error of zero for the CR, and then the MAE and % of eyes within +/- 0.25, 0.5, 0.75, and 1.0 D can be calculated and compared to the results from the machine learning models.
2. It should be discussed that long eyes over 26.3 mm and extreme long eyes were not included in this study. As shown in table 1, eyes with axial length up to of 26.26 mm were included in this study.
3. Both eyes of some patients were enrolled. It would be ideal to use one eye from each subject.
4. Multiple statistical comparison issue should be addressed.
5. If it is not statistically different, wording of comparable or equivalent should be used. Words of “insignificantly better or worse” should be avoided.
6. Line 206: “The interrelationship between Rxpost a IOLimplanted…”, should this be “…between Rxpost AND IOLimplanted…”?

Reviewer 2 ·

Basic reporting

- There are some grammatical and typos errors, the authors should revise it carefully.
- Fig. 2,3,4,5 should be combined together (or to become two figures)
- The network architecture in Figure 6 is not clear and it did not show fully structure of their network.
- To mention the RBF network, authors should cite more recent publications, especially in the biomedical field, such as https://doi.org/10.1186/s12859-016-1369-y and https://doi.org/10.1186/s12859-016-1163-x
- Similar to SVM, it has been successfully used in the recent publications in biomedical informatics, such as https://doi.org/10.1016/j.ab.2019.02.017
- The authors must provide their source code for reproducing.

Experimental design

- The is a need to provide independent testing in their model because overfitting is one of the most concerns for this type of problem. Overfitting needs to be addressed and discussed.
- Authors use two networks (SVM-RM and MLNN-EM), but the performance results are not much different between these two. Why did they need to use both? I suggest that the authors could combine these two networks to become a hybrid network and it should perform better than each network alone.
- The authors should report the range of parameters when they performed hyperparameters optimization.
- Hyperparameters for MLNN need to be reported clearer.
- I wonder the authors should display training MSE and validation MSE in the same graph in Figure 7. It will help readers and reviewer evaluate their model.
- When will the neurons stop to increase in the MSE?

Validity of the findings

- The authors only compared with Clinical Results (CR) is not enough, via CR is only a baseline method. Authors must compare their model with the other models using similar approaches.
- As CNN has been used successfully in different fields, especially in biomedical data such as https://doi.org/10.1016/j.ab.2018.06.011, https://doi.org/10.1002/jcc.24842, or https://doi.org/10.7717/peerj-cs.177, the authors should compare their models with a CNN model. Otherwise, the authors have to discuss this as a further direction.

---

## Round 0.2 · accepted · Accept

I believe that all major comments have been addressed, the manuscript can now be published.

Reviewer 2 ·

Basic reporting

No comment.

Experimental design

No comment.

Validity of the findings

No comment.

Additional comments

No comment.